# Full-duplex reflective beamsteering metasurface featuring magnetless nonreciprocal amplification

Sajjad Taravati [1✉] & George V. Eleftheriades[1]

Nonreciprocal radiation refers to electromagnetic wave radiation in which a structure provides different responses under the change of the direction of the incident field. Modern wireless telecommunication systems demand versatile apparatuses which are capable of full-duplex nonreciprocal wave processing and amplification, especially in the reflective state. To realize such a functionality, we propose an architecture in which a chain of series cascaded radiating patches are integrated with nonreciprocal phase shifters, providing an original and efficient apparatus for full-duplex reflective beamsteering. Such an ultrathin reflective metasurface can provide directive and diverse radiation beams, large wave amplification, steerable beams by simply changing the bias of the gradient active nonmagnetic nonreciprocal phase shifters, and is immune to undesired time harmonics. Having accomplished all these functionalities in the reflective state, the metasurface represents a conspicuous apparatus for efficient, controllable and programmable wave engineering.

---

[1] The Edward S. Rogers Sr. Department of Electrical and Computer Engineering, University of Toronto, Toronto, ON, Canada. ✉email: sajjad.taravati@utoronto.ca

Over the past decade, the advent and development of metasurfaces has led to significant advances to wave processing in modern telecommunication and optical systems[1–14]. However, conventional static metasurfaces are restricted by the Lorentz reciprocity theorem, which significantly limits their applications as versatile wave shapers and wave processors in modern wireless communication systems. Ferrite-based magnetic materials have been used for nonreciprocity implementation but they are cumbersome, costly, are not compatible with printed circuit board technology, and are not suitable for high frequency applications and future generation telecommunication systems. It is an object of this study to overcome some of the above-noted disadvantages. Lately, there has been a substantial scientific attraction to magnetless active metasurfaces and metamaterials for nonreciprocal wave processing[8,12,14–21]. Magnet-free nonreciprocal metasurfaces provide huge degrees of freedom for arbitrary alteration of the wavevector and temporal frequency of electromagnetic waves[8,12,14–16,18–26]. These may be classified into two main categories, that is, space-time metasurfaces[12,18,20,20,21,25,27–29] and transistor-loaded metasurfaces[8,15,30–35]. Among these non-reciprocity approaches, transistor-based nonreciprocity is of high interest thanks to its immense capability for efficient nonreciprocal electromagnetic-wave amplification while breaking time reversal symmetry.

Active metasurfaces provide large degrees of freedom for arbitrary and unidirectional alteration of the wavevector and amplitude of electromagnetic waves[8,12,16,18,18,20,23–26,36]. They represent a class of compact dynamic wave processors for transmission of electromagnetic waves. Reflective active metasurfaces represent a class of metasurfaces for simple and advanced wave tailoring[18,20,25,27,31,32,34,37]. They can be installed on a wall or inside a device such as a cell phone or a laptop to provide a diverse range of wave engineering. The advent of active metasurfaces has led to a revolution in beamsteering, while the main focus has been on transmissive beamsteering metasurface[38–40], given the complexities of beamsteering in the reflective state.

This paper proposes a low-profile reflective metasurface for nonreciprocal wave engineering and electromagnetic wave radiation control. The proposed reflective metasurface is capable of providing full-duplex unidirectional wave amplification and beam-steering. We introduce an original metasurface architecture in which a chain of series cascaded radiating patches are integrated with nonreciprocal phase shifters, providing an efficient mechanism for wave reception, unilateral signal amplification, nonmagnetic nonreciprocal phase shifting, and steerable wave reflection. Such a functionality has not been reported previously and is expected to find various applications in modern telecommunication systems. We provide the theory, simulation, and experimental results of full-duplex nonreciprocal-beam-steering and wave amplification of these reflective metasurfaces. Such metasurfaces may be placed inside home or work places, thereby amplifying, transforming and directing the radiation pattern of a source antenna or a received wave non-reciprocally, while providing different radiation beams for the reception and reflection states. The proposed metasurface is composed of chains of transistor-based nonreciprocal phase shifters interconnected to antenna elements. The metasurface is endowed with directive, diverse, and asymmetric reflection and reception radiation beams, and tunable beam shapes. Furthermore, these beams can be steered by changing the DC bias of the nonreciprocal phase shifters. Moreover, there are no undesired harmonics, yielding a high power efficiency with significant wave amplification, which is of paramount importance for practical applications such as point to point full-duplex communications[41–43].

## Results

**Operation principle and architecture.** Figure 1 shows the realization of the proposed reflective transistor-based metasurface. The metasurface is formed by a set of phase-gradient cascaded radiator-amplifier-phaser supercells. The metasurface comprises a dielectric layer sandwiched between two conductor layers. The bottom conductor layer acts as the ground plane of the patch antenna elements, and also includes the direct current (DC) signal path of the unilateral circuits. The top conductor layer includes patch antenna elements, transistors, and phase shifters. The dielectric layer separates the two conductor layers from each other. Each supercell is formed by a patch antenna element, a phase shifter and a unilateral circuit. When an electromagnetic wave is received at the surface of the metasurface, the metasurface reflects a wave having an identical frequency to the frequency of the received wave but towards a desired direction in space. The metasurface system comprises a dielectric layer interposed between two conductor layers. Each of the conductor layers is formed by a plurality of supercells embedded therein. Each supercell in the plurality of supercells comprises a microstrip patch radiator in electrical connection with a phase shifter and a unilateral transistor-based amplifier. The transistor radio frequency (RF) circuit includes two decoupling capacitors, and the DC biasing circuit of the transistor includes a choke inductor, two bypass capacitors and one biasing resistor. A DC signal biases the transistors to create a gradient non-reciprocal phase shift profile.

Figure 2 sketches the high-level architecture of the proposed full-duplex nonreciprocal reflective beamsteering metasurface and its application to an advanced full-duplex indoor wireless communication system. The metasurface thickness is subwave-length. In the forward problem, the incoming wave from the right side impinges on the metasurface under the angle of incidence $\theta_i^F$ which is inside the reception beam of the metasurface. The reception beam of the metasurface is governed by the gradient phase shifters in each supercell. Hence, the wave is received by the metasurface, acquires a power gain and is reflected at the desired angle of reflection $\theta_r^F$, instead of the specular reflection angle $-\theta_i^F$ as in conventional reciprocal surfaces. In contrast, in the backward problem, the incoming wave from the left side impinges on the metasurface under an angle of incidence which is outside the reception beam of the metasurface. Therefore, the wave is not received by the metasurface and is reflected without significant reflection gain or with loss. To demonstrate full-duplex (nonreciprocal reflection) operation of the metasurface, as shown in Fig. 2, we consider spatial inversion of the time-reversed forward problem, i.e., $\theta_i^B = \theta_r^F$ [25]. For a reciprocal surface, the backward reflected wave should reflect under an angle of reflection which is equal to the forward incidence angle. However, given the nonreciprocal nature of the proposed metasurface,

$$\theta_r^B \neq \theta_i^F, \quad \text{and} \quad E_r^B \neq E_r^F. \quad (1)$$

Figure 3 describes the beamsteering mechanism, including wave incidence and reflection from the cascaded supercells of the metasurface in Figs. 1 and 2. Each chain is constituted by $N$ interconnected supercells, each of which is formed by a radiating patch element characterized by the length $L$ and phase shift $\phi_p$, a unilateral transistor-based amplifier characterized by the complex transmission function $T_U$ and the phase shift $\phi_U$, and a gradient phase shifter characterized by the complex transmission function $T_{\phi_n}$, and phase shift $\phi_n$, where $1 < n < N$. The incoming wave from the right side impinges on the metasurface under the angle of incidence $\theta_i$, which is inside the reception beam of the metasurface governed by the gradient phase shifters in each supercell. The

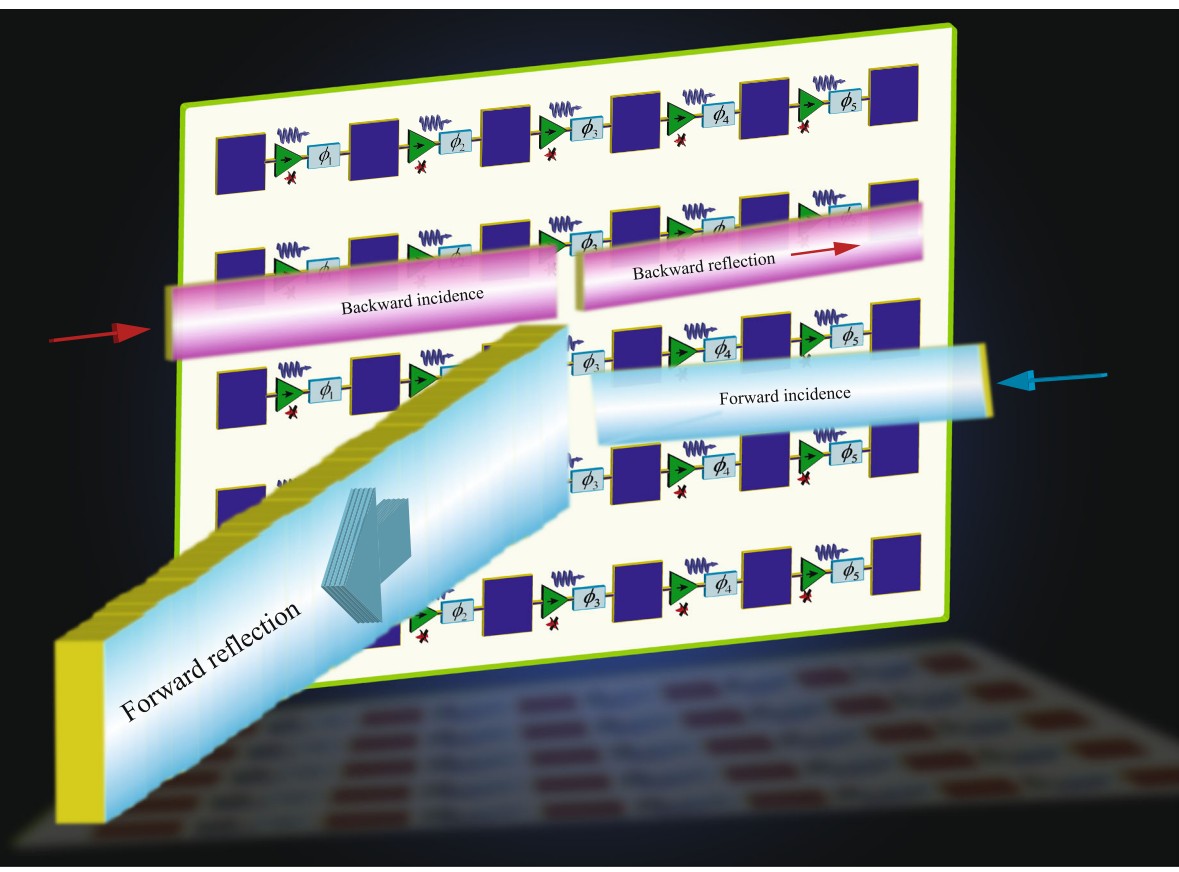

**Fig. 1 Schematic of the reflective beamsteering metasurface.** The metasurface is constituted of cascaded radiator-amplifier-phaser chains for simultaneous reflection and reception at different angles of reception and reflection.

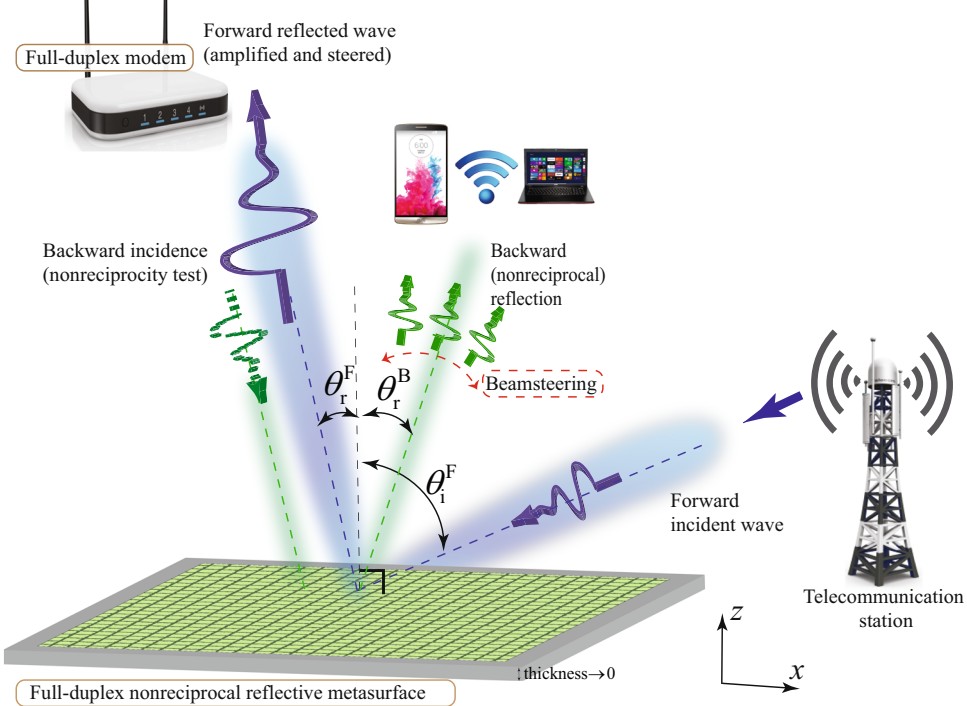

**Fig. 2 Metasurface functionality.** Full-duplex nonreciprocal reflective beamsteering and its application to an advanced full-duplex indoor wireless communication system.

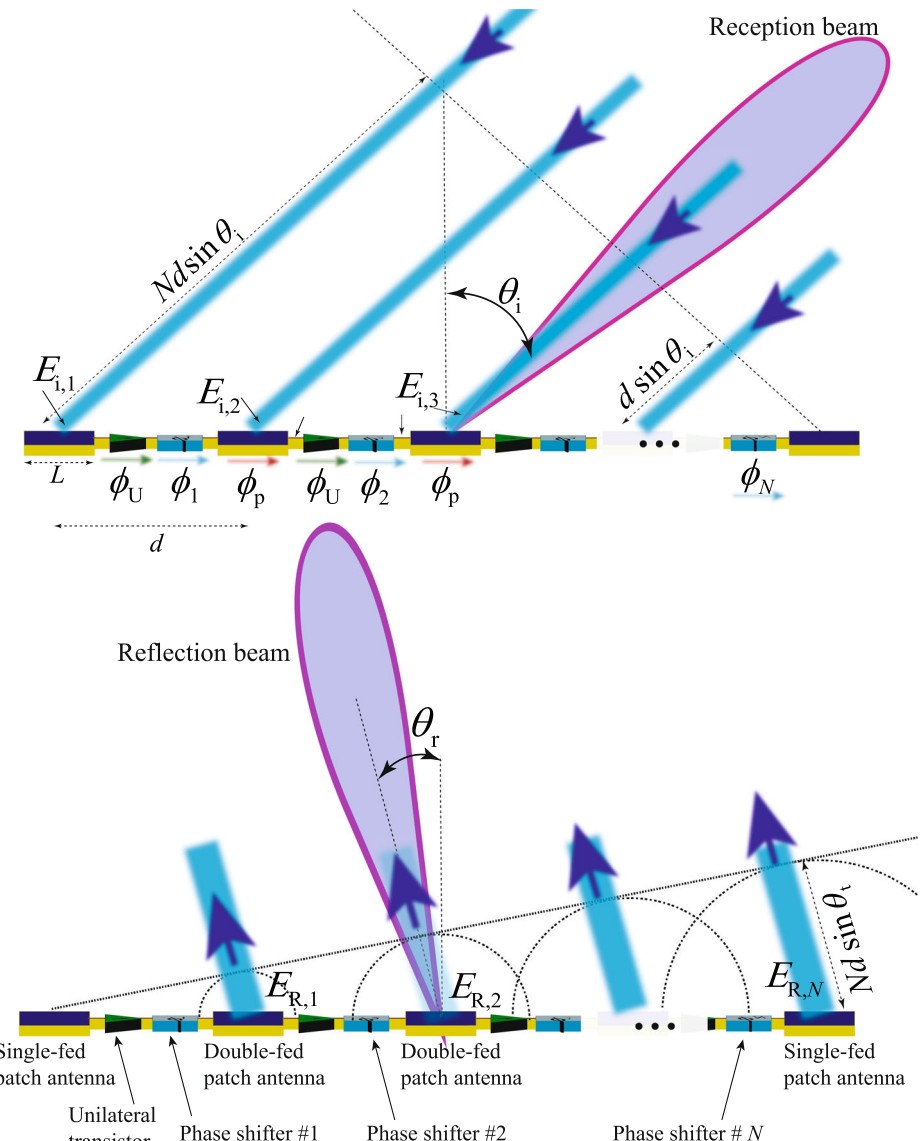

**Fig. 3 Beamsteering mechanism of the nonreciprocal reflective metasurface.** A chain of series cascaded radiating patches are integrated with nonmagnetic nonreciprocal phase shifters, providing an efficient mechanism for wave reception, one-way signal amplification, nonmagnetic nonreciprocal phase shifting, and steerable wave reflection.

incident wave is received by the radiating patch elements upon different phases corresponding to the angle of incidence $\theta_i$. Then, the received signal by each patch radiator may be written in terms of electric fields as

$$E_{i,n} = E_{0,n} \exp(i\beta(N-n)d\sin(\theta_i)), \qquad (2)$$

where $\beta$ is the wavenumber of the incident wave, and $d$ denotes the distance between two adjacent elements.

Figure 4a depicts a schematic of a chain of the interconnected supercells. Each supercell is composed of a patch antenna element, and a nonreciprocal phase shifter. The nonreciprocal phase shifters can be either one-way or two-way. A one-way nonreciprocal phase shifter is constituted of a unilateral device, e.g., a unilateral transistor-based amplifier, incorporated with a fixed phase shifter. The patch antenna elements are double-fed microstrip patch antennas to allow the flow of the reflected power in the desired direction inside the metasurface. However, the first and last patch antenna elements are single-fed patches. The chain of the interconnected patches and nonreciprocal phase shifters

behave differently with the incident waves from the right side and the left side.

Figure 4b illustrates the mechanism of iterative wave incidence and reflection from the interconnected patch radiators in the proposed active chain. Depending on the angle of incidence, the incoming wave to each supercell (shown by magenta arrows) may or may not be in phase with the traveling wave inside each supercell of the chain. A maximum gain assumes that the incoming waves are in phase with the traveling wave inside the chain. Hence, in case of any phase difference, the total gain will reduce. We shall stress than the radiation loss $T_{R,n}$ introduced by each patch radiator is very well desired and represents the operation principle (for beamsteering and nonreciprocity purposes) of this metasurface as shown in Fig. 4 a. Each chain comprises $N$ supercells, themselves composed of a unilateral component with the complex transfer function $T_{U,n}$, a phase shifter with the complex transfer function $T_{\phi,n}$ and a patch radiator with the complex radiation transmission function $T_{R,n}$ and complex transmission function of $T_{p,n}$ between its two feed lines, i.e., $T_{R,n} = 1 - T_{p,n}$. As a result, each supercell introduces a total inside-chain complex transmission function of $G_{T,n} = T_{U,n}T_{\phi,n}T_{p,n}$, and a

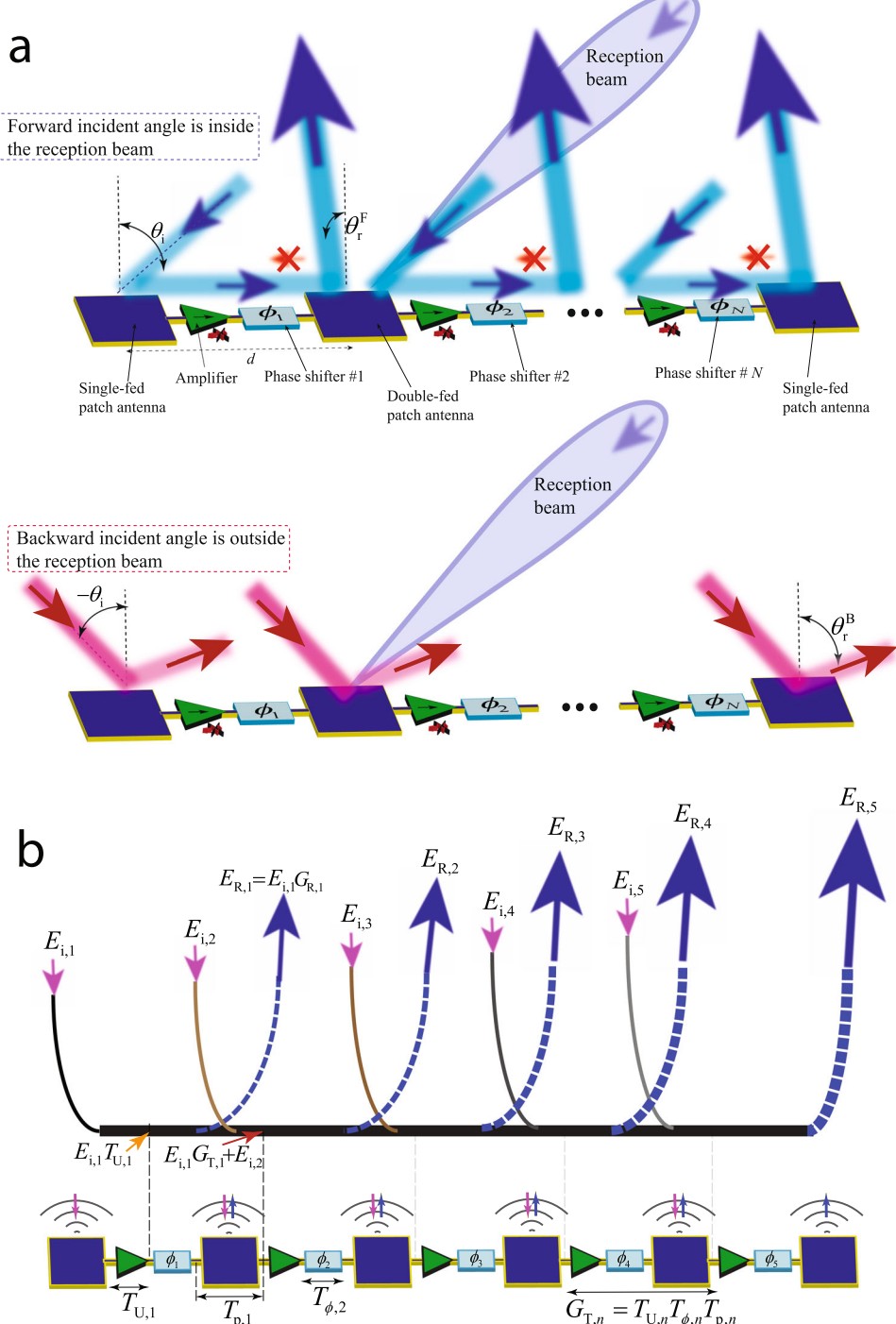

**Fig. 4 Operation principle. a** Nonreciprocity based on asymmetric reception beam. **b** Iterative wave incidence and reflection from the supercells of a chain and computation of the reflection angle and total gain using Eqs. (3) and (5), respectively.

complex radiation transmission function of $G_{R,n} = T_{U,n} T_{\phi,n} T_{R,n}$. Then, as shown in Fig. 4b, the electric field of the reflected wave from the $n$th supercell reads:

$$E_{R,n} = \left( E_{i,n} + \sum_{k=1}^{n-1} E_{i,k} G_{T,k} \right) G_{R,n}. \qquad (3)$$

The total power gain of each chain is equal to the average power gain of $n$ supercells inside a chain, i.e.,

$$G_{ch} = \frac{P_{out-ch}}{P_{in-ch}} = \frac{\sum_{n=1}^{N} |E_{R,n}|^2}{\sum_{n=1}^{N} |E_{i,n}|^2}, \qquad (4)$$

and the total power gain of an array system is equal to the average gain of $M$ chains, as

$$G_{tot} = \frac{P_{out}}{P_{in}} = \frac{1}{M} \sum_{m=1}^{M} G_{ch,m}, \qquad (5)$$

where $M$ is the number of the chains of the array.

**Experimental demonstration.** Figure 5a illustrates the detailed architecture of the fabricated reflective beamsteering metasurface, and Fig. 5b shows a photo of the fabricated reflective metasurface. Figure 5c illustrates the measurement set up for the nonreciprocal

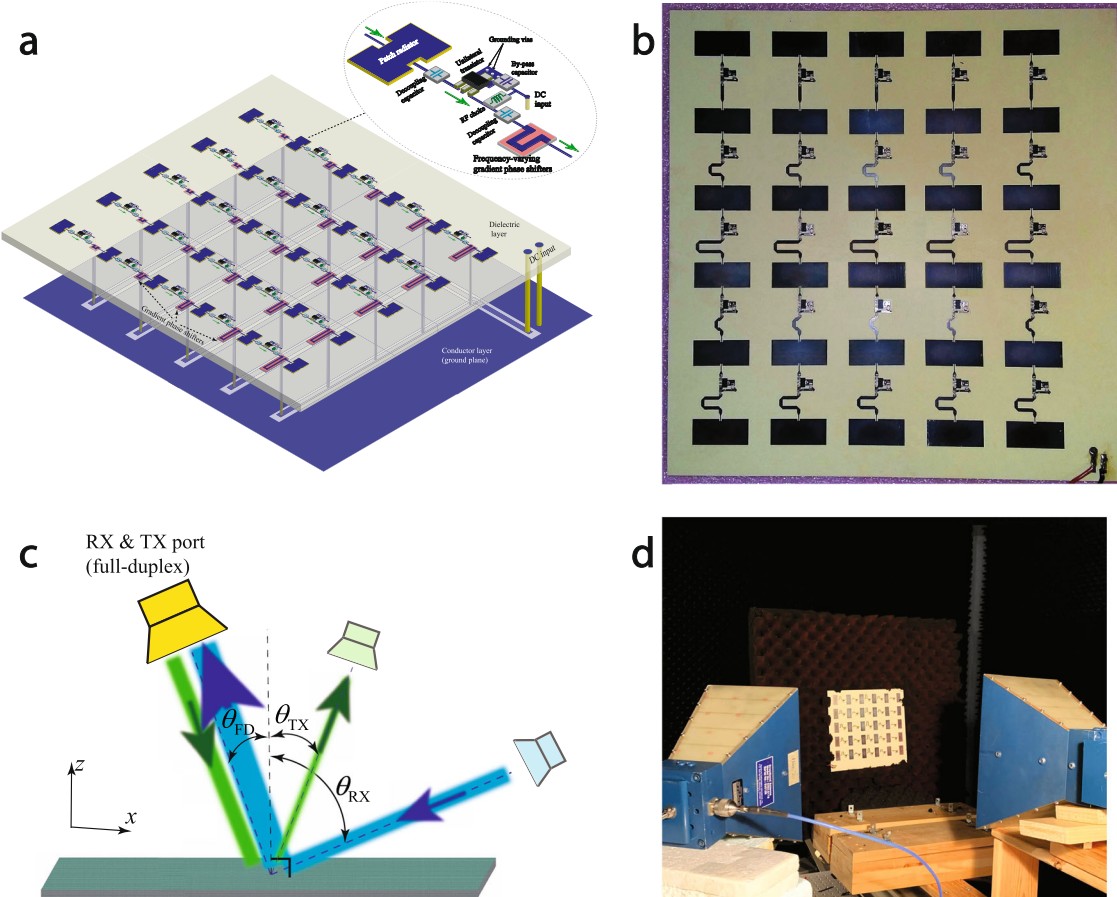

**Fig. 5 Experimental demonstration. a** Architecture of the fabricated reflective beamsteering metasurface. **b** A photo of the fabricated metasurface. **c**, **d** Measurement set-up.

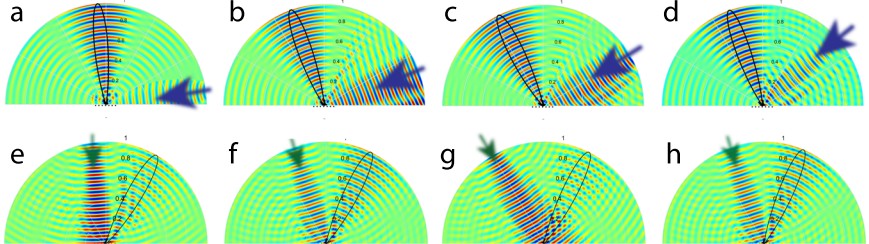

**Fig. 6 Full-wave simulation results for nonreciprocal reflective beamsteering at 5.8 GHz for a–d forward wave incidence, and e–h backward wave incidence for nonreciprocity examination. a** $\theta_i = 80°$. **b** $\theta_i = 70°$. **c** $\theta_i = 60°$. **d** $\theta_i = 50°$. **e** $\theta_i = -5°$. **f** $\theta_i = -20°$. **g** $\theta_i = -28.5°$. **h** $\theta_i = -20°$.

and full-duplex operation measurement of the proposed reflective nonreciprocal-beam metasurface. The nonreciprocity examination of such a reflective metasurface is as follows. The blue incoming wave from the right side impinges on the metasurface (under the angle of reception $\theta_{RX}$) is being amplified and reflected to the left side of the metasurface (under the angle of reception $\theta_{FD}$), where is being received by the yellow horn antennas. Next, to examine the nonreciprocity (full-duplex) of the structure, we illuminate the structure with the spatial inversion of the time-reversed of the blue path. Hence, the green (backward) wave is launched by the yellow horn antenna and impinges on the metasurface from the left side (under the angle $\theta_{FD}$), and is being reflected to the right side of the metasurface (under the angle of transmission $\theta_{TX}$). We notice that the angle of transmission is different than the angle of reception, i.e., $\theta_{TX} \neq \theta_{RX}$. A detailed explanation on nonreciprocity and asymmetry in electromagnetic

systems and their difference is described in ref. [25]. Figure 5d shows a photo of the measurement set-up. The measurement set-up consists of the fabricated reflective metasurface, an absorber for holding the metasurface, a vector network analyzer (VNA), a DC power supply and two horn antennas. We first accomplished a short-open-load-through (SOLT) calibration of the VNA across the frequency band. Next, we placed a perfect electric conductor (PEC) full-reflector inside the absorber (instead of the metasurface) and measured the free-space path loss (FSPL) between two horn antenna for the reflection path. Then we placed the metasurface and measured the results for different angles of incidence and reflection, which includes FSPL. Finally, we subtracted the metasurface plus FSPL results from the FSPL achieved from the PEC full-reflector experiment.

Figure 6a–h plot the full-wave simulation results illustrating the nonreciprocal reflective beamsteering mechanism of the

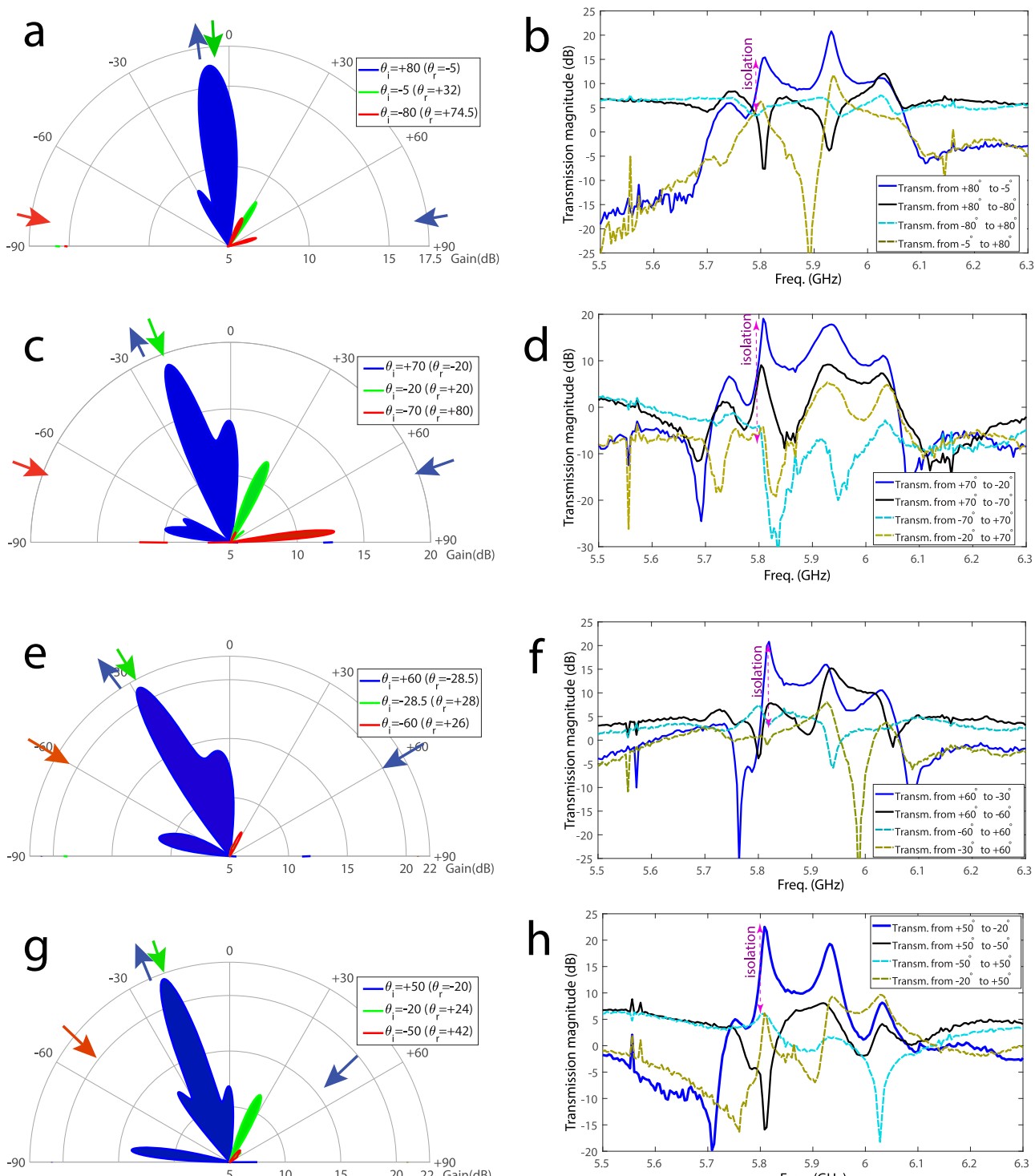

**Fig. 7 Experimental results for full-duplex reflective beamsteering.** The angular and frequency responses are measured for wave incidence upon different angles of incidence. Polar plots are measured at 5.81 GHz. **a**, **b** $\theta_i = 80°$. **c**, **d** $\theta_i = 70°$. **e**, **f** $\theta_i = 60°$. **g**, **h** $\theta_i = 50°$.

proposed metasurface. Here, the reflection gain of the metasurface for forward wave incidence is set to 1 for the sake of presentation so that both incident and reflected waves can be seen. As we see in these figures, the metasurface introduces different angles of reflection for each angle of incidence.

Figures 7a–h plot the experimental results demonstrating the full-duplex beam steering functionality of the metasurface for wave incidence from different angles of incidence, ranging from +80 to +50°. For the forward problem, where the incident blue

wave impinges on the metasurface from the right side, the wave is being amplified by the metasurface instantly and is reflected to the desired angle of reflection to be received by the full-duplex port (specified in Fig. 5c). For the backward problem, we examine both nonreciprocity (full-duplex operation) and asymmetry of the metasurface. The backward wave for nonreciprocity examination is demonstrated by the green arrow (backward incidence) in the left side of the polar figures (i.e., Fig. 7a, c, e, g) and green (backward) reflected beams in the right side of the same polar

**Table 1 Full-duplex nonreciprocal reflective beamsteering at 5.81 GHz.**

| Experiment number | 1 | 2 | 3 | 4 | 5 | 6 |
|---|---|---|---|---|---|---|
| Forward incidence angle | +40° | +45° | +50° | +60° | +70° | +80° |
| Forward reflection angle | 0° | −18° | −20° | −28.5° | −20° | −5° |
| Backward incidence angle | −40° | −45° | −50° | −60° | −70° | −80° |
| Backward reflection angle | 0° | +66° | +42° | +26° | +80° | +74.5° |
| Isolation level (dB) | >19 | >22 | >15 | >21 | >6 | >10 |
| Experimental forward gain (dB) | >21.5 | >25 | >21.5 | >21 | >19 | >16 |
| Theoretical forward gain (dB) | 21.1 | 25.8 | 21.9 | 21.6 | 18.7 | 16.4 |

figures. Furthermore, the backward wave for asymmetry examination is demonstrated by the red arrow (backward incidence) in the left side of the polar figures, and the red (backward) reflected beams in the right side of same polar figures. The polar plots in Fig. 7a, c, e, g are measured at 5.81 GHz. These polar plots demonstrate that the metasurface provides a unique full-duplex wave operation, and unidirectional amplification as a result of its nonreciprocal and asymmetric wave reflection. To further clarify the metasurface operation, we have plotted the frequency response of the metasurface for different angles of incidence and reflection in Fig. 7b, d, f, h.

The black solid line and cyan dashed line in the rectangular frequency response plots (i.e., Fig. 7b, d, f, h) examine the nonreciprocity at the specular angle of the metasurface. This metasurface is designed to offer its best performance for the angle of incidence of +50 (Fig. 7g, h), and as can be seen it introduces more than 20 dB isolation at the specular angle, i.e., introduces about 5 dB transmission gain from −50° to +50°, and less than −15 dB from +50° to −50°. In addition, the main beam of the reflected beam is at −20° with 21.5 dB transmission gain, which is more than 36 dB stronger than the reflected wave at the specular angle of −50°. We shall stress that for all four angles of incidence, i.e., +80°, +70°, +60°, and +50°, the reflection at the specular angle is 10 dB below the main beam of the metasurface. This shows the flexibility of the metasurface, so that the metasurface chain would be designed according to the given specifications, i.e., angle of incidence, main beam gain, nonreciprocity, specular angle isolation, half-power beam width (HPBW) of the main beam, etc.

Table 1 lists a summary of the full-duplex nonreciprocal beamsteering reflective metasurface performance. The theoretical results (computed using Eqs. (2–5)) and experimental results in Table 1 show that the metasurface provides more than 16 dB gain for all angles of incidence. The amplifiers are supplied by a DC voltage of 3.84 V and DC current of 42 mA. As a result, the power efficiency of the metasurface, i.e., efficiency = $(P_{out} − P_{in})/P_{DC}$ is equal to 77.6% (for 21 dB gain in Fig. 7e, g). Such a power efficiency range is affected by several factors, including the angle of incidence, efficiency of the transistor-based amplifiers, the radiation pattern of the patch radiators, the transfer function of the gradient phase shifters, and the iterative amplification of portion of the traveling wave inside the chain.

Next, we show the strong nonreciprocal amplification regime of the metasurface, where $\theta_i < 45°$. Here, more than 21 dB reflection gain for forward wave incidence is achieved while the backward reflection gain is less than 3 dB. The metasurface is designed to present full amplification for $\theta_i = 40°$ corresponding to the normal reflection, i.e., $\theta_r = 0°$. Figure 8a plots the experimental results demonstrating the nonreciprocal full-duplex wave amplification functionality for wave incidence from the angle of incidence of 40°. For the forward problem, where the incident wave impinges on the metasurface from the right side, i.e., upon the angle of incidence of +40°, the wave is being amplified, about 21.6 dB, by the metasurface instantly and is

reflected to the desired angle of reflection of zero degree. However, for the backward problem, where the incident wave impinges on the metasurface from the left side, i.e., under the angle of incidence of −40°, the wave is not amplified significantly.

Figure 8b plots the experimental results demonstrating the nonreciprocal full-duplex beam steering functionality for wave incidence from the angle of incidence of 45°. For the forward problem, where the incident wave impinges on the metasurface from the right side, i.e., upon the angle of incidence of +45°, the wave is being amplified, more than 25 dB, by the metasurface instantly and is reflected to the desired angle of reflection of −18°. However, for the backward problem, where the incident wave impinges on the metasurface from the left side, i.e., under the angle of incidence of −45°, the wave is not amplified significantly and is not beam-steered.

Figure 9a plots the experimental results demonstrating the beam steering functionality through changing the phase shift of the nonreciprocal phase shifters for wave incidence from the angle of incidence of +30° at the frequency 5.8 GHz. For the forward problem, where the incident wave impinges on the metasurface from the right side, i.e., upon the angle of incidence of +60°, the wave is being amplified more than 10 dB by the metasurface instantly and is reflected to different desired angles of reflection for the DC bias of 3.7 and 3.84 V.

Figure 9b plots the experimental results demonstrating the beam steering functionality through changing the phase shift of the nonreciprocal phase shifters, by the DC bias, for wave incidence from the angle of incidence of +60° at 5.8 GHz. For the forward problem, where the incident wave impinges on the metasurface from the right side, i.e., upon the angle of incidence of +60°, the wave is being amplified more than 10 dB, by the metasurface instantly and is reflected to different desired angles of reflection for the DC bias of 3.6, 3.84, and 4 V.

Figure 10a shows a schematic representation of the near-field experimental set-up of the nonreciprocal radiation beam reflective metasurface. In this experiment, the two source horn antennas are placed inside the near-field zone of the metasurface and very close to the metasurface. Figure 10b plots the experimental results demonstrating the near-field performance of the metasurface for wave incidence from the angle of incidence of +40°. This figure shows that the metasurface provides very close results for both far-field and near-field experiments. This shows great performance of the metasurface in the near-field. Such a unique near-field performance, i.e., near-field wave amplification, nonreciprocity, and beam-steering, is expected to find numerous applications in 6G indoor wireless communications.

## Discussion

The main concept of this paper is the realization of a reflective metasurface, which is capable of nonreciprocal beam generation. Such a metasurface realizes full-duplex nonreciprocal-beam-steering and amplification in the reflective state, where

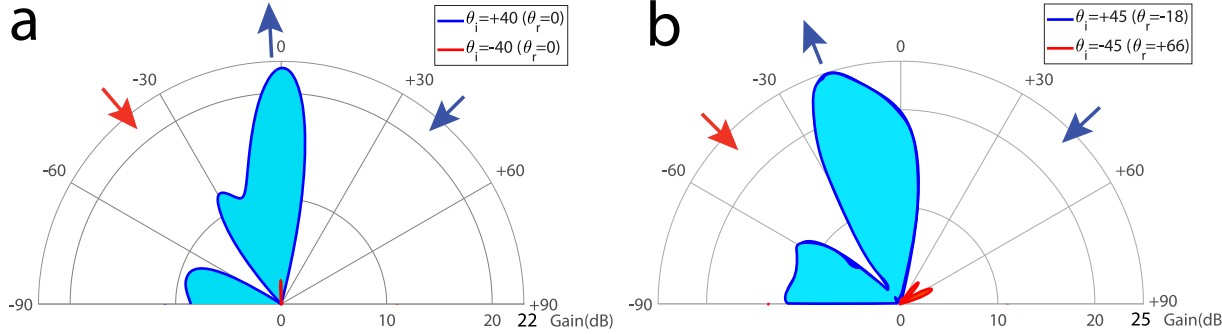

**Fig. 8 Experimental results for nonreciprocal wave amplification at 5.81 GHz. a** $\theta_i = 40°$. **b** $\theta_i = 45°$.

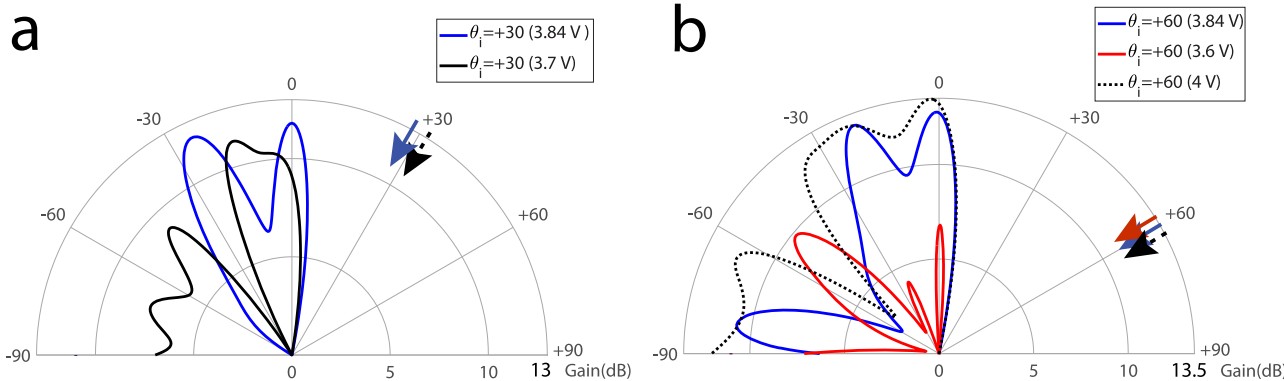

**Fig. 9 Experimental results for the controllable beamsteering mechanism. a** $\theta_i = 30°$ at 5.81 GHz. **b** $\theta_i = 60°$ at 5.81 GHz.

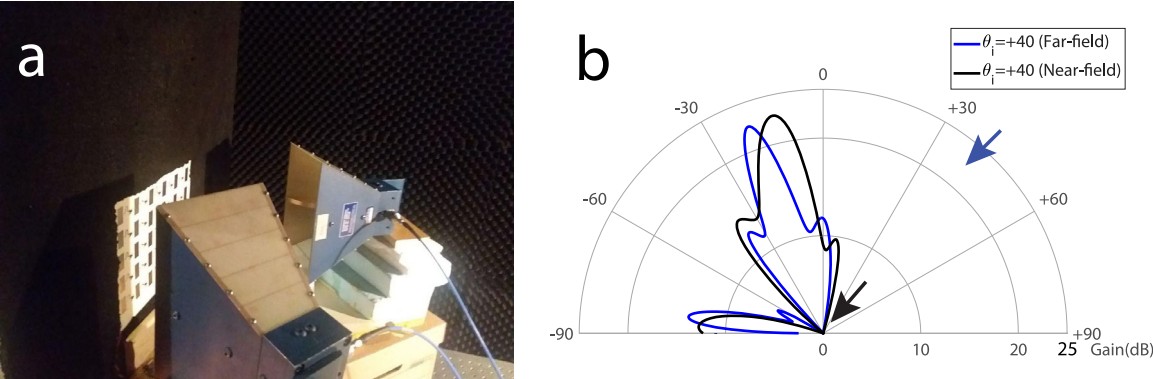

**Fig. 10 Experimental results for near-field efficiency of the reflective metasurface at 5.81 GHz. a** A photo of the near-field experimental set-up. **b** Near-field beam versus far-field beam of the metasurface for wave incidence upon the angle of incidence $\theta_i = 40°$.

simultaneous reception and reflection of waves is accomplished but at different reflection angles, devoid of any undesired frequency conversion and with distinct reception and reflection gains. This is totally different than other proposed nonreciprocal metasurfaces[16,20,27,35,44,45] in which the metasurface changes the spectrum of the incident wave, and introduces an undesired frequency alteration. The recently proposed nonreciprocal reflective time-modulated metasurfaces[16,20,27,35,45] suffer from an unwanted frequency conversion in the spectrum of the incident wave, so that the reflected wave acquires a different frequency than the incident wave. Such a frequency change is very impractical as the frequency conversion ratio is very small so that one cannot achieve a practical frequency conversion functionality. However, in our proposed nonreciprocal-beam metasurface, the

incident and reflected waves share the same frequency. Hence, the proposed metasurface is more practical. Furthermore, some of the recently proposed nonreciprocal metasurfaces are transmissive structures[8,12], which are not suitable for practical applications. In contrast, the proposed reflective metasurface in this study is very practical as it can be mounted on a wall and provide a desired beamsteering and amplification.

We have introduced an architecture for reflective wave engineering comprising chains of patch antenna elements with embedded non-reciprocal amplifying phase shifters. This architecture is unique even in the case of traditional reciprocal reflect-arrays[46,47]. For the proposed non-reciprocal reflective surface, there is no inherent limit to the bandwidth as the frequency bandwidth of the proposed unit cells can be easily enhanced

through engineering approaches for the bandwidth enhancement of patch resonators[48,49].

In terms of applications, such metasurfaces can be elegantly mounted on a wall or on a smart device in a seamless way. These surfaces are capable of massive MIMO beam-forming, as no excessive RF feed lines and matching circuits are required, the metasurface functionality and operation can be fully controlled and programmed through biasing of unilateral devices and phase shifters, as well as tunable patch radiators. Highly directive and reflective full-duplex nonreciprocal-beam operation is a very promising feature of the proposed metasurface to be used for a low-cost high capability and programmable wireless beam-forming. The metasurfaces can become the core of an intelligent connectivity solution for signal enhancement in WiFi, cellular, satellite receivers, and IoT sensors. It provides fast scanning between users while providing full-duplex multiple access and signal coding.

## Methods

### Fabricated metasurface architecture and maximum achievable gain.
The metasurface comprises a dielectric layer sandwiched between two conductor layers, formed by an array chain of supercells. Supercells are composed of patch antenna elements and non-reciprocal tunable phase shifters. When an electromagnetic wave at a given frequency impinges on the metasurface, the metasurface amplifies the wave instantly and reflects the wave to a desired direction, having an identical frequency to the frequency of the incident wave. The feeding line supporting the DC bias of the unilateral amplifiers is embedded inside the bottom conductor ground-plane layer. The specifications of the supercells may be varied via a DC signal to control the properties of the reflected wave, including the angle of reflection and the amplitude of the reflected wave. Each supercell comprises one reciprocal phase shifter, one unilateral transistor-based amplifier, one choke inductance, two decoupling capacitances, and one bypass capacitor. The choke inductance prevents leakage of the incident electromagnetic wave to the DC biasing path, and the decoupling capacitances prevent leakage of the DC bias to the RF path of the next supercell. Each chain comprises five supercells, themselves composed of a unilateral component with about 12 dB gain ($T_{U,n} = 15.85$), a phase shifter with about 1 dB insertion loss ($T_{\phi,n} \approx 0.794$) and a patch radiator with about 6 dB radiation loss ($T_{p,n} = 0.25$ and $T_{R,n} = 0.75$). As a result, each supercell introduces 4.95 dB total transmission gain ($G_{T,n} = 3.13$), yielding a maximum gain of 25.8 dB, according to the theoretical results computed using Eqs. (2–5), and experimental results in Fig. 7a–h and Table 1.

### Fabricated prototype details.
The metasurface is fabricated as a two-layer circuit, i.e., two conductor layers and one dielectric layer, made of Rogers RO4350 with the dielectric constant $\epsilon_r = 3.66$, dissipation factor $\tan \delta = 0.0037$, and 30 mils height. The metasurface is formed by 30 patch antenna elements, i.e., 20 double-fed and 10 single-fed patch antenna elements, and 25 nonreciprocal phase shifters. The top layer includes a set of chains of patches interconnected through one-way transistor-based gradient nonreciprocal phase shifters. The bottom conductor layer includes two metallic sheets, one acting as the RF grounding of the patch antennas, and the other one provides the DC bias of the transistors. The DC bias of the transistors is supplied to the bottom-right side of the top layer, transferred to the bottom layer through a via hole, and then supplied to each transistor through a via hole. Each nonreciprocal phase shifter includes a reciprocal transmission-line based phase shifter, a GAli-2+ transistor-based unilateral amplifier, two decoupling capacitors, an inductor and a bypass capacitor. A total number of 25 Gali-2+ unilateral transistor-based amplifiers, 25 choke inductors of 15 nH, 25 bypass capacitors of 100 pF, and 50 decoupling capacitances of 3 pF are used.

### Full-wave simulation.
Supplementary Fig. 1a, b show the simulation results, using electromagnetic field simulation software CST Studio Suite, for the power flow and power loss inside a passive chain formed by six interconnected patch radiators, respectively. Our simulation results show that each transmitting double-fed patch introduces more than 6 dB radiation loss between its two feed lines, so that about 75% of the power injected to the left feed-line of the patch is re-radiated to air and the remained 25% of the power is transmitted to the feed-line of the next right-side patch. For this purpose, the local incident field on a patch is provided by a source patch above the leftmost patch. This incoming wave impinges on the 1st far-left patch (reception state patch), which induces a current to its feeding port on its right-hand side. Then, this signal is delivered to the 2nd to 6th patches on the right-hand side of the receiving (1st) patch. Subsequently, considering 6 dB radiation loss of each patch radiator, the 2nd patch re-radiates 75% of the power to the air, while the remaining 25% of the power is delivered to the 3rd to 6th patches. These two figures show strong reception of the incoming wave by the leftmost patch antenna, and strong flow of power to the second patch radiator and strong re-radiation (transmission to air) by the second patch radiator. However, as most of the power is re-radiated by the second patch, far less power flows through the

third patch. Supplementary Fig. 1c shows the power flow on top of the inter-connected chain structure. Downward arrows in the left side show reception of the incoming wave by the leftmost patch, red-upward arrows in the second patch illustrate strong transmission (re-radiation) through the second patch radiator, and green-upward arrows on the third patch indicate weak transmission (re-radiation) through the third and its right-side patches.

### Realization based on bilateral nonreciprocal phase shifter.
For further development of the proposed metasurface and for achieving a more versatile structure, one may use a two-way nonreciprocal phase shifter and amplifier as illustrated in (Supplementary Fig. 2). Such a nonreciprocal phase shifter is formed by two power dividers, two unilateral transistor-based amplifiers, two fixed phase shifters, and four decoupling capacitors. The top and bottom phase shifters provide different phase shifts. The top and bottom amplifiers may provide equal amplification and isolation, in the forward and backward directions, respectively. The signal entering the structure from the left side goes through the upper arm, experiences amplification by the top amplifier and then passes through the top phase shifter. However, the signal entering the structure from the right side goes through the lower arm, experiences amplification by the bottom amplifier and then passes through the bottom phase shifter.

## Data availability
The data that support the findings of this study are available from the corresponding author upon reasonable request.

## Code availability
The codes that are used to generate results in the paper are available from the corresponding author upon reasonable request.

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

## Acknowledgements
This work was supported in part by TandemLaunch Inc. and LATYS Intelligence, Montreal, QC, Canada, and in part by the Natural Sciences and Engineering Research Council of Canada (NSERC). The authors would like to especially thank Mr. Gursimran Singh Sethi, Co-founder and Technical Leader of LATYS Intelligence, and Dr. Omar Zahr, Director of Technology at TandemLaunch Inc., for their great help and support.

## Author contributions
S.T. carried out the analytical modeling, numerical simulations, sample fabrication, and measurements. G.V.E. planned, coordinated, and supervised the work. All authors discussed the theoretical and experimental aspects and interpreted the results. All authors contributed to the preparation and writing of the manuscript.

## Competing interests
The authors declare no competing interests.
