## [Peer Review File · Nature Communications]

Reviewers' comments:

Reviewer #1 (Remarks to the Author):

This paper proposed a full-duplex reflective beamsteering metasurface design featuring magnetless nonreciprocal amplification. Some problems should be addressed.

(1) Fig. 7(a) shows that these gain block amplifiers are cascaded with radiating patches and time delay lines. This gain block has almost over 12 dB gain at this frequency from its datasheet. If five gain block amplifiers cascaded with each other, the cascaded gain should be over more than 60 dB. I think there could have three reasons for this low gain of the cascaded structure, which are

(a) Radiating patch element has very low radiation efficiency.

(b) The return loss at the gain block amplifier is very poor. Because this gain block amplifier is internally matched to 50 Ohms.

(c) The incident wave illuminates each element which has very large path loss.

The authors should clarify this issue.

(2) At the end of Section 1 (introduction), the authors claim "the proposed metasurface has no undesired harmonics, yielding a high conversion efficiency with significant wave amplification". However, for this architecture the conversion efficiency is not high because most of power of gain block amplifier might be not work well. The authors should explain the definition of conversion efficiency. Moreover, in the following sections, the authors fail to discuss the conversion efficiency based on experiments. I encourage the author to talk about this important aspect in the text.

(3) In Section 2 (Operation principle), the received signal at the feeding ports of the patch elements is expressed as Eq. (3). The reviewer considers the sum symbol Σ could not express the single feeding port of patch element. Further, in Eq. (4) and (5), the E-field expressions are incorrect because left dimension is V/m while right dimension is V. Besides, Eq. (4) and (5) are lack of space propagation factor, please correct these equations.

(4) In Section 2 (Operation principle), the authors claim "For further development of the proposed metasurface and for achieving a more versatile structure, one may use a two-way nonreciprocal phase shifter and amplifier". However, power amplifier could not work at this architecture for a full-duplex application because there may have loop gain by poor two-way isolation and return loss of output and input port to make power amplifier self-oscillation, unless just one power amplifier work at time such as TDD application. And for TDD application this architecture which is shown on Fig.5 is not a full-duplex circuit. The authors should clarify this issue.

(5) The manuscript's format is not coincidence with the Nature Communication standard, which should include introduction, results, discussion and etc.

Reviewer #2 (Remarks to the Author):

The authors report on the design of nonreciprocal beam-steering metasurfaces with transistor-based unit cells. The patch antennas inside the unit cells are designed in such a way to ensure the flow of the reflected power in the desired unilateral direction inside the metasurface. Overall, I found this manuscript scientifically correct and reporting interesting findings. I believe that it could attract certain attention among the metasurface as well as 6G telecommunication communities.

However, I am not convinced that this manuscript provides sufficient level of impact for Nature Communications. The concept of nonreciprocal beam steering is not novel and has been explored in multiple works [2,17,19,21,23,25-29], and also in [Nonreciprocal Phased-Array Antennas].

It is true that in the current manuscript a novel approach towards nonreciprocal beam steering was suggested, however, it does not make the concept itself new. Moreover, the approach for obtaining nonreciprocity (not specifically beam steering) using transistor-based metasurfaces was known too. The only

advantage that I see is the absence of frequency side bands in the present metasurface compared to time-modulated systems, which is an advancement for the application side but is not a new physical effect. Hence, I believe the paper will be more suitable for more specialized electrical engineering journals (Nature Electronics, IEEE TAP, Advanced Functional Materials, etc.).

Based on these points, regretfully, I cannot recommend the manuscript for publication. Below I attach a list of comments that could improve quality of the manuscript:

1. Please reduce the length of the abstract to make it concise and straight to point. Some information is repeated twice there. For example, statements on wireless telecommunications.
2. The introduction is messy and should be significantly improved. Among others, I list several problems. Several sentences are repeated: "The proposed apparatus may be placed on a wall or in front of an antenna to amplify a wave, and/or steer a beam to a desired direction, i.e., transform the radiation pattern and introduce..."

The order of sentences is sometimes unlogical: "Such a unique, extraordinary and useful functionality has not been reported previously and is expected to find various applications in modern telecommunication systems. We provide the theory, simulation and experimental results of full-duplex nonreciprocal-beam-steering and wave amplification of these reflective metasurfaces. Such metasurfaces may be placed on a wall, thereby amplifying, transforming and directing the radiation pattern of a source antenna or a received wave non-reciprocally, while providing different radiation beams for the reception and reflection states". Moreover, the authors should engage the reader by properly explaining such important concepts as full-duplex communication and nonreciprocity. These concepts are not necessarily familiar to the broad audience of Nat. Comm.

3. "They may be classified into two main categories, that is, space-time metasurfaces [12,17,19,21,21,23,25–29] and transistor-loaded metasurfaces [8,15,30–32,32–34]".

Please refer to the earlier works on transistor-loaded metasurfaces:

[Artificial Faraday rotation using a ring metamaterial structure without static magnetic field], [Electromagnetic modeling of a magnetless nonreciprocal gyrotropic metasurface], [Nonreciprocal magnetless CRLH leaky-wave antenna based on a ring metamaterial structure].

4. It would be useful to provide some references for beam steering in metasurfaces.
5. Does the choke inductor include ferrite? The presence of the ferrite would conflict with the "magnet-free" labeling. It makes sense to comment on this minor issue for preciseness.
6. Please provide dielectric constant of Rogers substrate used for reader's convenience.
7. Please comment on how high frequencies the present metasurface approach can work at. Is there an upper limit due to electronics speed?
8. At what frequency figs. 10 (a,c,e,g) are plotted? Please add this information in the caption.
9. Please use different sets of colors in the left and right subfigures of fig. 10. One may expect that same color corresponds to the same illumination scenario, why it is not the case.
10. I would suggest to drastically shorten the text description of the results in figs. 10-11 since all the results are nicely summarized in Table 1. It is very complicated to read all this information with many numbers in the text. Moreover, the text and conclusions for different scenario are repeating. Instead, it would be beneficial to summarize in the text the results of Table 1, comparing the performance at different angles. This is now missing.
11. In this work, nonreciprocity level is measured as the transmission difference between two horn antennas when they are in the original positions and when their positions are interchanged. While the reviewer agrees with this approach for defining and measuring nonreciprocity level, this choice was not explained in the text. Please elaborate on this and provide some references in support of this approach (preferably some classical books).
12. Please comment on whether the nonreciprocal metasurface will operate under simultaneous illumination by two waves from different directions.
13. I think it is better to remove Fig. 1 or combine it with Fig. 2. What is the purpose of showing the topology of the metasurface in several figures.
14. Why in the left subfigures of fig. 10 the authors show only blue output arrows, but not red and green?

Reviewer #3 (Remarks to the Author):

This manuscript proposed a full-duplex reflective metasurface with nonreciprocal beam steering and wave amplification. The metasurface architecture consists of chains of patch antenna elements with embedded non-reciprocal amplifying phase shifters. It is an interesting work for wireless communication to solve the important duplex functionalities. However, this manuscript is far from mature for publication in the Nature Communication. I am worried about whether the paper contains enough scientific novelty compared to previous active non-reciprocal metasurfaces and whether their results can support the conclusion. Additionally, the paper is written in poor quality. My detailed technical comments are listed below.

1. The explanation of "full-duplex" is unclear. Is it identical to the conventional "full-duplex" in present industrial applications? It is not easy to understand the "full-duplex" phenomenon through the experiment for audience among various communities.
2. There are too many figures in the manuscript. Please carefully read the guideline for authors. Too many similar pictures are duplicated in the manuscript, such as Figure 1 and Figure 2.
3. The abstract was written in a complicated and unclear way. The academic issue behind the critical industrial bottleneck should be stressed. Authors claimed the potential application for several times. I cannot understand why. However, the key parameters and performance of proposed metasurface were not provided at all. Frequency? Isolation? Gain? Steering status? And others.
4. In the introduction, the author mentioned a lot related work of active metasurfaces with non-reciprocal transistor-loaded architecture, which looks very similar to author's proposed one. Can authors compare their work to previous literatures and show their difference and breakthrough? The reference should be checked carefully. For example, in Row 34, and 35, "...of the wavevector and temporal frequency of electromagnetic waves 8,12,14-23,23-25 34, there are two ref. 23. And there are others. Furthermore, authors once again claim how their work would be useful with several paragraphs. It looks boring. The introduction part should be rewritten as well as the abstract.
5. In Figure 4, how can we understand the full-duplex? Please explain it in the main body and provide necessary notation in figure caption. Also, Fig.4 and Fig.3 present the same content, which can be merged into one.
6. In Row 88, regarding "under the angle of incidence θ_i which is inside the reception beam of the metasurface", the angle range were not given in the paper.
7. Figure 5 and the related description are not highly related to the main topic of the article. They can be moved to Supplementary Materials.
8. Figures. 6-8 can be put together. Detailed information of experiment setup should be provided. I cannot find them in either main body or methods. All the experiments should be carried out in the farfield region. However, from the figures, it looks the distance between horn antenna and the metasurface is very short, no matter for the cases of the so-called "nearfield" or "farfield" in the manuscript.
9. In Fig. 10, the black lines have no corresponding definition and explanation in the paper. Please compare the proposed isolation with present industrial state-of-art. How did authors measure isolation in open free space? Please provide equations.
10. For the description of figure10, there are too much repetitive contents. The sentences are almost the same, while only parameters are different. It can be written in a concise way. Additionally, there is no need to separate Fig.11 from Fig.10.
11. To support the "Programmable and Controllable Beam steering", further experiments should be conducted.

12.The discussion and conclusion were highly repetitive. Personally, I strongly suggest that the same content should be mentioned in the paper only once. The paper should be comprehensive but concise.

13.What does the notation (1,2, to 7) mean in table 1? It looks meaning less.

Response to the Reviewer's comments.

We would like to thank the Editorial Board Member and the Reviewers for their time and comments, which helped to improve the quality of our manuscript.

This document uses the following color code:

- **Green:** original comments of the reviewer.
- **Black:** response of the authors.
- **Blue:** changes introduced to the paper.

All the numbers of the figures and references in the following refer to the original manuscript unless otherwise specified.

Reviewers' comments:

Reviewer #1 (Remarks to the Author):

This paper proposed a full-duplex reflective beamsteering metasurface design featuring magnetless nonreciprocal amplification. Some problems should be addressed.

(1) Fig. 7(a) shows that these gain block amplifiers are cascaded with radiating patches and time delay lines. This gain block has almost over 12 dB gain at this frequency from its datasheet. If five gain block amplifiers cascaded with each other, the cascaded gain should be over more than 60 dB. I think there could have three reasons for this low gain of the cascaded structure, which are

(a) Radiating patch element has very low radiation efficiency.

(b) The return loss at the gain block amplifier is very poor. Because this gain block amplifier is internally matched to 50 Ohms.

(c) The incident wave illuminates each element which has very large path loss.

The authors should clarify this issue.

We believe that the reviewer has made a factual error, and his/her comment represents misunderstanding of our structure. To further clarify this, we have prepared a model for the gain calculation of each chain, as shown in Fig. R1 below. This model provides the details of the iterative transmission and reception by each chain and the associated gain. Each chain comprises 5 supercells, themselves composed of a gain block with about 12 dB gain, a phase shifter with about 1 dB insertion loss and a patch radiator with about 6 dB "radiation loss". As a result, each

supercell introduces 4.95 dB (3.13) total transmission gain, yielding a **maximum gain of 25.8 dB** for each chain through averaging the five reflected beams (and as we measured in Fig. 11 of our paper).

```

GR=9.38;
G=3.13;
G1=GR;
G2=GR*(1+G);
G3=GR*(1+G+G^2);
G4=GR*(1+G+G^2+G^3);
G5=GR*(1+G+G^2+G^3+G^4);
Gav=10*log10((G1+G2+G3+G4+G5)/5)

```

Gav =

25.8333

Fig. R1. A model for computing the gain of the proposed non-reciprocal chain.

Depending on the angle of incidence, the 5 incoming beams (shown by magenta arrows in Fig. R1) may or may not be in phase with the traveling wave inside the chain. The maximum gain of

25.8dB assumes that these 5 incoming waves are all in phase with the traveling wave inside the chain. In case there is any phase difference, the total gain will reduce.

We shall stress that the (6 dB) radiation loss introduced by each patch radiator is very well desired and represents the operation principle (for beamsteering and nonreciprocity purposes) of this metasurface as shown in Fig. 4(top) of our original submission.

Additionally, we have simulated a small chain which includes three patches (without any amplifiers or phase shifters). Figures R2 and R3 below show the power flow inside and on top of the chain. For this purpose, the local incident field on a patch is provided by a ‘source’ patch above the leftmost patch. This incoming wave impinges on the 1st far-left patch (reception state patch) which induces a current to its feeding port on its right-hand side. Then, this signal is delivered to the 2nd and 3rd patches on the right-hand side of the receiving (1st) patch. Then, considering 6 dB radiation loss of each patch radiator, the 2nd (middle) patch re-radiates 75% of the power to the air, while 25% the power is delivered to the 3rd patch (rightmost). Once more, this supports the conceptual picture of Fig. 4 (top) in our original paper and shows that there is no strong attenuation between adjacent patches as the reviewer erroneously concluded. In addition, these simulations are consistent with our gain analysis shown previously.

Fig. R2. Power flow inside the interconnected chain structure, showing strong reception of the incoming wave by the leftmost patch antenna and strong flow and transmission of power to the middle patch antenna. However, as most of the power is re-radiated by the middle patch, far less power flows through the rightmost patch.

Fig. R3. Power flow on top of the interconnected chain structure. Downward arrows in the left side show reception of the incoming wave by the leftmost patch, red-upward arrows in the middle illustrate strong transmission (re-radiation) through the middle patch antenna, and green-upward arrows on the right indicate weak transmission (re-radiation) through the rightmost patch.

(2) At the end of Section 1 (introduction), the authors claim “the proposed metasurface has no undesired harmonics, yielding a high conversion efficiency with significant wave amplification”. However, for this architecture the conversion efficiency is not high because most of power of gain block amplifier might be not work well. The authors should explain the definition of conversion efficiency. Moreover, in the following sections, the authors fail to discuss the conversion efficiency based on experiments. I encourage the author to talk about this important aspect in the text.

We have added this explanation to the paper, which comes after calling Table 1. Furthermore, Equation (6) clarifies the expected power gain of the array metasurface.

(3) In Section 2 (Operation principle), the received signal at the feeding ports of the patch elements is expressed as Eq. (3). The reviewer considers the sum symbol \sum could not express the single feeding port of patch element.

Equation (3) represents the total received voltages at the input of all patches not for a single patch, as it was clearly written inside text before the equation.

Further, in Eq. (4) and (5), the E-field expressions are incorrect because left dimension is V/m while right dimension is V. Besides, Eq. (4) and (5) are lack of space propagation factor, please correct these equations.

Equations (4) and (5) have been modified.

(4) In Section 2 (Operation principle), the authors claim “For further development of the proposed metasurface and for achieving a more versatile structure, one may use a two-way nonreciprocal phase shifter and amplifier”. However, power amplifier could not work at this architecture for a full-duplex application because there may have loop gain by poor two-way isolation and return loss of output and input port to make power amplifier self-oscillation, unless just one power amplifier work at time such as TDD application. And for TDD application this architecture which is shown on Fig.5 is not a full-duplex circuit. The authors should clarify this issue.

The proposed alternative nonreciprocal phase shifter may provide more degrees of freedom for controlling different features of the reflective metasurface as follows. In general, the two amplifiers (for forward and backward directions) of the structure are biased at different points and provide different transmission gains and phases. As a consequence, in contrast to the current unidirectional nonreciprocal phase shifter in Fig. 4 (composed of only one amplifier and one phase shifter) which suppresses the backward wave strongly, the bi-directional nonreciprocal phase shifter allows a two-way transmission but with different transmission “magnitudes” and phases. As a result of this “unequal” two-way transmission, a full-duplex operation is still achievable in a more versatile and controllable manner, e.g., for controlling the shape of the reflected beam (HPBW, etc.).

(5) The manuscript’s format is not coincidence with the Nature Communication standard, which should include introduction, results, discussion and etc.

This has been modified.

Reviewer #2 (Remarks to the Author):

The authors report on the design of nonreciprocal beam-steering metasurfaces with transistor-based unit cells. The patch antennas inside the unit cells are designed in such a way to ensure the flow of the reflected power in the desired unilateral direction inside the metasurface. Overall, I found this manuscript scientifically correct and reporting interesting findings. I believe that it could attract certain attention among the metasurface as well as 6G telecommunication communities.

However, I am not convinced that this manuscript provides sufficient level of impact for Nature

Communications. The concept of nonreciprocal beam steering is not novel and has been explored in multiple works [2,17,19,21,23,25-29], and also in [Nonreciprocal Phased-Array Antennas].

It is true that in the current manuscript a novel approach towards nonreciprocal beam steering was suggested, however, it does not make the concept itself new. Moreover, the approach for obtaining nonreciprocity (not specifically beam steering) using transistor-based metasurfaces was known too. The only advantage that I see is the absence of frequency side bands in the present metasurface compared to time-modulated systems, which is an advancement for the application side but is not a new physical effect. Hence, I believe the paper will be more suitable for more specialized electrical engineering journals (Nature Electronics, IEEE TAP, Advanced Functional Materials, etc.).

We respectfully disagree with the reviewer. Unfortunately, the reviewer seems to have failed to appreciate the unique architecture and operating principle of our work. In this way, it was not possible to distinguish the differences between our proposed reflective metasurface with previously reported structures.

For instance, the reference mentioned by the reviewer, [Nonreciprocal Phased-Array Antennas], is about a nonreciprocal **time-modulated antenna**, suffering from **undesired time harmonic** sidebands and **transmission loss**. We shall stress that the mentioned structure is an **antenna and not a reflective metasurface**. Furthermore, it is suffering from an undesired and impractical frequency conversion. Hence, such an antenna is completely different than our proposed structure.

In contrast, our proposed metasurface is organized based on an original architecture offering features that are not seen in previously reported metasurfaces and antennas, that is, full-duplex transmission in a reflective metasurface, beamsteering, and nonreciprocal amplification in the reflective mode. Figure R4 below shows how the proposed full-duplex reflective beamsteering and amplification can lead to an unprecedented and unique functionality that will attract both academic and commercial interests.

1. Please reduce the length of the abstract to make it concise and straight to point. Some information is repeated twice there. For example, statements on wireless telecommunications.

We have shortened the abstract and removed the unnecessary parts.

Fig. R4. Application of the proposed nonreciprocal reflective beamsteering metasurface to an advanced full-duplex indoor wireless communication system.

2. The introduction is messy and should be significantly improved. Among others, I list several problems. Several sentences are repeated: "The proposed apparatus may be placed on a wall or in front of an antenna to amplify a wave, and/or steer a beam to a desired direction, i.e., transform the radiation pattern and introduce..."

The order of sentences is sometimes unlogical: "Such a unique, extraordinary and useful functionality has not been reported previously and is expected to find various applications in modern telecommunication systems. We provide the theory, simulation and experimental results of full-duplex nonreciprocal-beam-steering and wave amplification of these reflective metasurfaces. Such metasurfaces may be placed on a wall, thereby amplifying, transforming and directing the radiation pattern of a source antenna or a received wave non-reciprocally, while providing different radiation beams for the reception and reflection states".

We have updated the abstract and introduction to ensure that there is no repetition of content.

Moreover, the authors should engage the reader by properly explaining such important concepts as full-duplex communication and nonreciprocity. These concepts are not necessarily familiar to the broad audience of Nat. Comm.

To further clarify the full-duplex operation, especially in the reflective state, we have updated Fig. 2.

3. “They may be classified into two main categories, that is, space-time metasurfaces [12,17,19,21,21,23,25–29] and transistor-loaded metasurfaces [8,15,30–32,32–34]”.

Please refer to the earlier works on transistor-loaded metasurfaces:

[r1]-Artificial Faraday rotation using a ring metamaterial structure without static magnetic field,

[r2]-Electromagnetic modeling of a magnetless nonreciprocal gyrotropic metasurface,

[r3]-Nonreciprocal magnetless CRLH leaky-wave antenna based on a ring metamaterial structure.

Reference [r3] **is not a metasurface**, rather a leaky-wave antenna; hence the comparison with our work is misguided. Furthermore, the two references, [r1], [r2], are similar structures: That is these are metasurfaces using transistor-loaded “scatterers”. Perhaps the most relevant paper is the one of using transistor-loaded loops to emulate Faraday rotation [r1]. This and our work both use transistors to emulate non reciprocity. But this is where the similarity ends. The way that our reflecting metasurface is organized (see the response to reviewer#1 above), its operation and its full-duplex capability are a big step forward to the state of the art. Please refer to Fig. R4 above which further highlights the full-duplex operation of our structure in a realistic wireless telecommunication scenario. This should set it well apart from all cited references by this reviewer.

4. It would be useful to provide some references for beam steering in metasurfaces.

Proper references and explanation have been added to the introduction.

5. Does the choke inductor include ferrite? The presence of the ferrite would conflict with the “magnet-free” labeling. It makes sense to comment on this minor issue for preciseness.

No, we have used air-core inductors.

6. Please provide dielectric constant of Rogers substrate used for reader’s convenience.

This information has been added to the paper.

7. Please comment on how high frequencies the present metasurface approach can work at. Is there an upper limit due to electronics speed?

Such metasurfaces are suitable for microwave and millimeter wave frequencies, where transistors are available. In addition, the proposed metasurface may as well be adopted for terahertz applications, where for instance graphene-based transistors are utilized.

8. At what frequency figs. 10 (a,c,e,g) are plotted? Please add this information in the caption.

This information has been added to the caption.

9. Please use different sets of colors in the left and right subfigures of fig. 10. One may expect that same color corresponds to the same illumination scenario, why it is not the case.

This has been modified.

10. I would suggest to drastically shorten the text description of the results in figs. 10-11 since all the results are nicely summarized in Table 1. It is very complicated to read all this information with many numbers in the text. Moreover, the text and conclusions for different scenarios are repeating. Instead, it would be beneficial to summarize in the text the results of Table 1, comparing the performance at different angles. This is now missing.

We have updated the text describing Fig. 10 (Fig. 9 of the revised paper) and summarized the text.

11. In this work, nonreciprocity level is measured as the transmission difference between two horn antennas when they are in the original positions and when their positions are interchanged. While the reviewer agrees with this approach for defining and measuring nonreciprocity level, this choice was not explained in the text. Please elaborate on this and provide some references in support of this approach (preferably some classical books).

This information has been added to the paper, where explaining Fig. 7(c), and proper reference has been introduced.

12. Please comment on whether the nonreciprocal metasurface will operate under simultaneous illumination by two waves from different directions.

Yes, the metasurface operates under simultaneous illumination by two waves from different directions, and this represents the full-duplex operation of the metasurface. This scenario is clarified in Fig. 2 of the revised paper, where the metasurface is illuminated from both right and left leading to two outputs at the opposite side.

13. I think it is better to remove Fig. 1 or combine it with Fig. 2. What is the purpose of showing the topology of the metasurface in several figures.

We have updated Fig. 2 to show the application of the proposed structure in wireless communication systems. Fig. 1 show the metasurface with its inclusions and nonreciprocal wave reflection from it. We believe they are both required for better understanding of the paper especially now that Fig. 2 shows the full-duplex application of the structure.

14. Why in the left subfigures of fig. 10 the authors show only blue output arrows, but not red and green?

All outputs for blue, green and red excitations are shown in Fig. 10 (Fig. 9 in the revised paper). However, due to the nonreciprocal reflection of the metasurface (which is desired), the outputs of green and red excitations are much weaker than the blue output, so that they are tiny or do not appear in the result. For instance, in Figs. 10(a),(c) and (g) the green output appears in the right side of plot but in Fig. 10(e), the green output is very weak and do not show up in the plot.

Reviewer #3 (Remarks to the Author):

This manuscript proposed a full-duplex reflective metasurface with nonreciprocal beam steering and wave amplification. The metasurface architecture consists of chains of patch antenna elements with embedded non-reciprocal amplifying phase shifters. It is an interesting work for wireless communication to solve the important duplex functionalities. However, this manuscript is far from mature for publication in the Nature Communication. I am worried about whether the paper contains enough scientific novelty compared to previous active non-reciprocal metasurfaces and whether their results can support the conclusion. Additionally, the paper is written in poor quality. My detailed technical comments are listed below.

1. The explanation of “full-duplex” is unclear. Is it identical to the conventional “full-duplex” in present industrial applications? It is not easy to understand the “full-duplex” phenomenon

through the experiment for audience among various communities.

The full-duplex operation has been clarified in Fig. 2. Full-duplex operation in the reflective state is indeed an original concept that has been introduced in this paper and represents a useful concept for efficient wireless communication, especially inside home and office spaces.

2. There are too many figures in the manuscript. Please carefully read the guideline for authors. Too many similar pictures are duplicated in the manuscript, such as Figure 1 and Figure 2.

We have removed the unnecessary figures. However, we believe that Figures 1 and 2 are different and necessary, especially now that we have modified Fig. 2 to highlight the full-duplex operation and application of the metasurface. Figure 1 shows the metasurface and its inclusions, e.g., different chains and the wave incidences, while Figure 2 focuses on the metasurface functionality, operation principle and application, asymmetric and nonreciprocal wave reflections.

3. The abstract was written in a complicated and unclear way. The academic issue behind the critical industrial bottleneck should be stressed. Authors claimed the potential application for several times. I cannot understand why. However, the key parameters and performance of proposed metasurface were not provided at all. Frequency? Isolation? Gain? Steering status? And others.

We have reformulated the abstract to ensure that there is no repetition. The detailed specifications of the proposed metasurface (Frequency, Isolation, Gain, Steering status) are provided inside the paper.

4. In the introduction, the author mentioned a lot related work of active metasurfaces with non-reciprocal transistor-loaded architecture, which looks very similar to author's proposed one. Can authors compare their work to previous literatures and show their difference and breakthrough? The reference should be checked carefully. For example, in Row 34, and 35, "...of the wavevector and temporal frequency of electromagnetic waves 8,12,14–23,23–25 34, there are two ref. 23. And there are others. Furthermore, authors once again claim how their work would be useful with several paragraphs. It looks boring. The introduction part should be rewritten as

well as the abstract.

We have updated the abstract and introduction.

5. In Figure 4, how can we understand the full-duplex? Please explain it in the main body and provide necessary notation in figure caption.

Full-duplex operation is clarified in Fig. 2 and in Fig. 7(c), where the metasurface supports reflection from right to left and from left to right but at different angles of reflection. We have added a text when calling Fig. 7(c) to clarify the scenario.

Also, Fig.4 and Fig.3 present the same content, which can be merged into one.

Figure 3 shows the array-based reception and reflection mechanism of the metasurface while Fig. 4 (Fig. 5 in the revised version) describes the nonreciprocal reflection mechanism of the structure.

6. In Row 88, regarding “under the angle of incidence θ_i which is inside the reception beam of the metasurface”, the angle range were not given in the paper.

We have added Equation (6) which shows the effect of the incoming waves to each patch element and their contribution to the traveling wave inside a chain. Equation (6) and Fig. 3 describe the effect of the incident angle on the reflected beam. Our experimental results show that for the designed prototype, the full-duplex operation is guaranteed for the angle of incidence from 50 degrees to 80 degrees. To change the angle of incidence range and the angle of reflection, the gradient phase shifters and the transfer function of unilateral components should be changed.

7. Figure 5 and the related description are not highly related to the main topic of the article. They can be moved to Supplementary Materials.

Figure 5 of the original paper and the related description have been transferred to the supplemental materials.

8. Figures. 6-8 can be put together. Detailed information of experiment setup should be provided. I cannot find them in either main body or methods. All the experiments should be carried out in the farfield region. However, from the figures, it looks the distance between horn antenna and the

metasurface is very short, no matter for the cases of the so-called “nearfield” or “farfield” in the manuscript.

We have put Figures 6-8 together. The distance between the horn antennas and the metasurface is 1.4m for far-field measurement and 0.28m for near-field measurement. We are interested in the near-field performance of the metasurface as one of the main applications of this reflective metasurface is for WLAN beamsteering and amplification purposes inside small areas like home and offices (on the wall).

9. In Fig. 10, the black lines have no corresponding definition and explanation in the paper. Please compare the proposed isolation with present industrial state-of-art. How did authors measure isolation in open free space? Please provide equations.

Black lines represent specular reflection level. We have explained this inside the text. In addition, we have added the details of the measurement inside the text in the description of Fig. 7(d).

10. For the description of figure 10, there are too much repetitive contents. The sentences are almost the same, while only parameters are different. It can be written in a concise way. Additionally, there is no need to separate Fig. 11 from Fig. 10.

We have updated the description of Fig. 10 and removed unnecessary text.

Figure 10 provides the results for full-duplex operation of the structure while Fig. 11 provides the results for nonreciprocal amplification operation of the metasurface.

11. To support the “Programmable and Controllable Beam steering”, further experiments should be conducted.

Figure 11 (of the revised paper) shows that the reflected beam of the fabricated proof-of-concept metasurface can be controlled by the DC bias of unilateral amplifiers. In addition, one can utilize variable gradient phase shifters to have full control over the beam of the metasurface. In the commercialization step, one can use an FPGA to digitally control the functionality of the metasurface.

12. The discussion and conclusion were highly repetitive. Personally, I strongly suggest that the

same content should be mentioned in the paper only once. The paper should be comprehensive but concise.

We have updated the discussion and conclusion to ensure that there is no repetition of similar contents.

13. What does the notation (1,2, to 7) mean in table 1? It looks meaning less.

This has been modified.

REVIEWERS' COMMENTS

Reviewer #1 (Remarks to the Author):

This paper proposed a new architecture in which a chain of series cascaded radiating patches integrated with nonreciprocal phase shifters. The revised manuscript gave a detailed reply and revision, which is recommended for publication.

Reviewer #2 (Remarks to the Author):

The authors have replied to all my comments and improved the manuscript. Now I am convinced that the paper can be published in this journal.

Reviewer #3 (Remarks to the Author):

Review #2 and I have similar concern on the scientific novelty. However, authors did not respond our concern properly. Indeed, the authors ignored my concern.

In response to review #2, the authors claim the difference between antennas and their reflective metasurface. Antennas are a class of electromagnetic devices for radiating/transmitting/receiving signals. Antennas could be source-free (passive) or active. Antennas could be reflection-type as well.

In contrast, metasurface is a physical concept based on generalized Snell's law, which can be used as the design methodology for various devices, including antennas. Authors totally confuse the concept of antennas and metasurface. The response "the mentioned structure is an antenna and not a reflective metasurface" is not suitable. Authors' proposed structure, of course, is a reflective metasurface, and moreover, is still a kind of reflection-type antenna, which has been reported in the previous literature.

Unfortunately, I still do not see any fundamental, conceptual novelty in this paper after reading authors' reply. My opinion does not change. This paper is not suitable for Nature Communications.

Response to the Reviewer's comment.

We would like to thank the Editor and the Reviewer for their time and comments.

This document uses the following color code:

- **Green:** original comments of the reviewer.
- **Black:** response of the authors.

Review #3

Review #2 and I have similar concern on the scientific novelty. However, authors did not respond our concern properly. Indeed, the authors ignored my concern.

In response to review #2, the authors claim the difference between antennas and their reflective metasurface. Antennas are a class of electromagnetic devices for radiating/transmitting/receiving signals. Antennas could be source-free (passive) or active. Antennas could be reflection-type as well.

In contrast, metasurface is a physical concept based on generalized Snell's law, which can be used as the design methodology for various devices, including antennas. Authors totally confuse the concept of antennas and metasurface. The response "the mentioned structure is an antenna and not a reflective metasurface" is not suitable. Authors' proposed structure, of course, is a reflective metasurface, and moreover, is still a kind of reflection-type antenna, which has been reported in the previous literature.

We stand by our statement that our structure is a reflective metasurface and not an antenna. A reflective metasurface translates an incoming space-wave to another desired radiating space-wave. In contrast, antennas are one-port devices that translate an incoming space-wave to a signal through the induced current at the port of the antenna. The operation principle and required definition of our proposed reflective metasurface has clearly been explained in our paper and there is no need for any further clarification.